# Chronology and Geochemistry of Early Cretaceous Magmatism in the Northwestern Erguna Block, Northeast China

**Yan Li [1,2], Jia-Rui Cui [2,*], Zhi-Bin Han [2], Feng-Jun Nie [1,*], Xiao-Gung Hou [2] and Zhao-Bin Yan [1]**

[1] State Key Laboratory of Nuclear Resource and Environment, East China University of Technology, Nanchang 330013, China; hgyliyan@163.com (Y.L.); yzbqw@126.com (Z.-B.Y.)

[2] Research Institute NO.240, The China National Nuclear Corporation, Shenyang 110032, China; hzb_240dzkj@163.com (Z.-B.H.); houxiaoguang88@163.com (X.-G.H.)

[*] Correspondence: cjr18524429867@163.com (J.-R.C.); niefj@263.net (F.-J.N.)

**Abstract:** This study was conducted to define the background structure and petrogenetic significance of the Early Cretaceous magmatic rocks in the Badaguan area of northern Daxing'anling and to explore the Late Paleozoic tectonic evolution of the Mongol-Okhotsk suture zone. The Early Cretaceous magmatic rock was systematically investigated using zircon U-Pb dating and geochemical and petrological analyses. The results show that the rock: mainly consists of granites and rhyolites; has an age of 125–140 Ma; has a strong MgO, $Al_2O_3$, and total alkali content; has a $SiO_2$ content of 61.68 wt% to 77.41 wt%; and contains Rb, Th, U, and light REEs with depleted levels of of Sr, P, Ti, and heavy REEs. When combined with the Hf isotopic characteristics of the Early Cretaceous magmatic rock from the Erguna Massif, these results suggest that the magma originated from the partial melting of basal crustal materials during the Neoproterozoic–Phanerozoic period and that various mineral forms (including hornblende, plagioclase, and apatite) underwent fractional crystallization processes during the evolution of the magma. The Early Cretaceous magmatic rock from the Badaguan area recorded the extensional environment of the lithosphere after the closure of the Mongol-Okhotsk Ocean, and this hypothesis is consistent with the results of previous studies on the tectono-magmatic activities in Northeast China during the same period.

**Keywords:** Early Cretaceous magmatic rock; Mongol-Okhotsk Ocean; petrogenesis; geochemistry; Badaguan

## 1. Introduction

Three countries surround the Xingmeng orogenic belt: Mongolia, eastern Russia, and the Inner Mongolia–Northeast China region [1–3]. This region is one of the important components of the Central Asian orogenic belt (CAOB) and thus, is the focus of global research. The Daxing'anling mountains are located in the eastern part of the Xingmeng orogenic belt and are known for their vast area of Mesozoic volcanic rock and complex Phanerozoic magmatic rock, which forms the world-renowned "volcanic belt" and "granite sea" [4,5]. Following the closure of the Paleo-Asian Ocean during the Paleozoic era, this region experienced a series of complex evolutionary processes, such as oceanic crustal subduction, collisional orogeny, and convergence and splicing, which eventually formed the current distribution of geological units [5–8]. From east to west, these geological units are as follows: the Jiamusi Massif, the Songliao Massif, the Xing'an Massif, and the Erguna Massif (Figure 1b) [4,9,10]. Previous research has focused on the following constructive domains of the Paleo-Asian Ocean: the tectonic evolution of the Paleo-Pacific Ocean and its impacts on Northeast China [11–13]. More research has been conducted on the evolution of the Mongol-Okhotsk tectonic system in recent years and Late Triassic–Early Jurassic magmatic rock and contemporaneous porphyry copper–molybdenum deposits have been discovered there, both of which are closely related to the subduction of the Mongol-Okhotsk Ocean [14–16]. However, these studies have mainly concentrated on the period of oceanic

crustal subduction, whereas the Cretaceous tectonic evolutionary process following the closure of the Mongol-Okhotsk Ocean has rarely been explored and has not yet been fully elucidated. Studies have suggested that the magmatic evolution during the Cretaceous period, following the closure of the Mongol-Okhotsk Ocean, was of great importance for Northeast China, particularly for the Daxing'anling region [15,17,18].

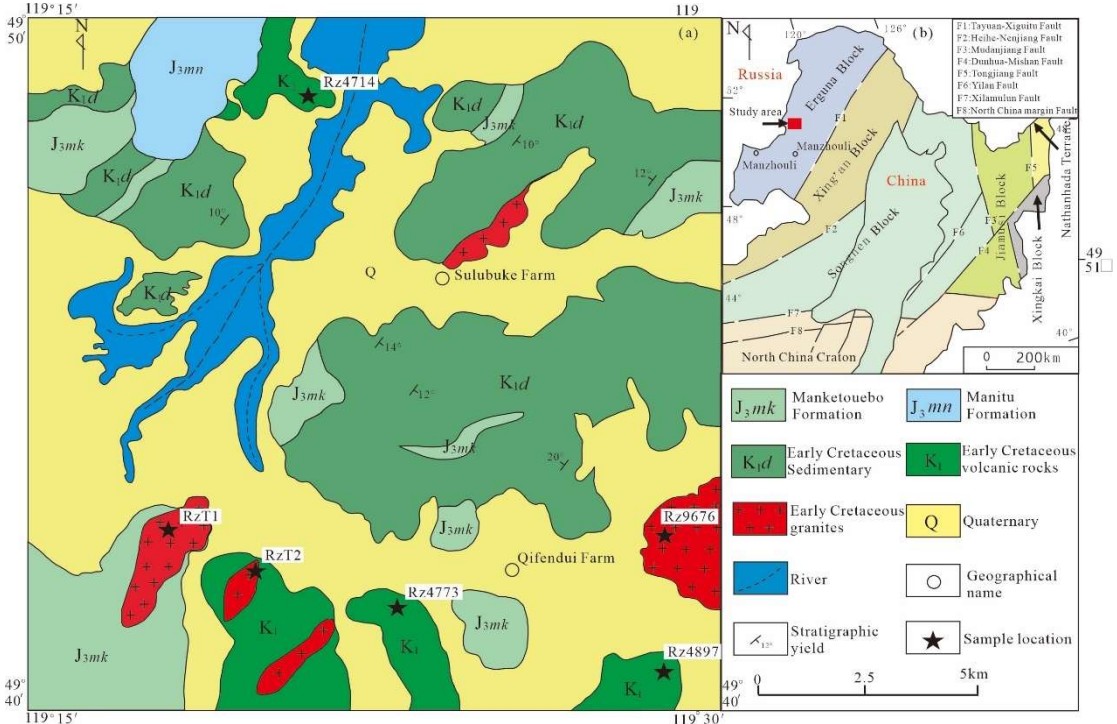

**Figure 1.** (**a**) A geological sketch map of NE China and (**b**) a geological sketch map of the Badaguan area (modified from Li et al. [7]).

The Erguna Massif is an important tectonic unit in the eastern part of the Central Asian orogenic belt in the northernmost part of Northeast China. The Tayuan–Xiguitu fault lies to its southeast and the Mongol-Okhotsk tectonic belt to its northwest (Figure 1).

The basement rock is mainly composed of Neoproterozoic metamorphic volcanic–sedimentary rock, igneous rock, and some Paleoproterozoic gneisses [19]. Granite is widely distributed over the Erguna Massif and it has been traditionally considered to have been formed during the Proterozoic–Paleozoic era [20,21]. However, subsequent research has found that the granites were actually formed in different periods, ranging widely from the Late Triassic to the Cretaceous [6,15,19]. Together with the Late Jurassic–Early Cretaceous volcanic rock, these granites were formed by the largest magmatism of the Erguna Massif as the strongest magmatic activity of the Early Cretaceous period occurred in the Erguna Block and its surrounding area [18]. The Erguna Block is, therefore, an excellent region for studying the features of the Mongol-Okhotsk orogenic belt as the existing magmatic rock provides a clear record of the history of its tectonic evolution. Late Mesozoic magmatic activities were strong and widely distributed in this area, and the geological formations from the lower to the upper levels consist of the Tamulangou Formation ($J_2t$), Manketou'ebo Formation($J_3mk$), Manitou Formation ($J_3mn$), and Baiyingaolao Formation ($K_1b$) [20,21]. The intrusive rock is mainly syenogranite, monzogranite, and granodiorite, which were primarily formed during the Late Triassic period [16].

This study was conducted to explore the zircon U-Pb ages and the whole-rock geochemistry of the Early Cretaceous magmatic rock in the Badaguan area of the northern section of Daxing'anling. The chronological, geochemical, and isotopic data obtained enabled an investigation of the petrogenesis of the Late Mesozoic magmatism in the Erguna Block. The study aimed to provide important evidence to further the exploration of

the tectonic evolution of Northeast China and beyond. The study area is located in the Badaguan area of the Inner Mongolia Autonomous Region to the north of Hailar City. Its geotectonic location makes it a part of the Erguna Massif (Figure 1b). Small amounts of the pre-Mesozoic strata are exposed in the study area, and only extremely small amounts of metamorphic rock from the Jiageda Formation are visible around the Ergun River.

The Early Cretaceous magmatic rock primarily consists of granite and rhyolite (Figure 2a,b) and is spread in a northeasterly direction. Granite intrudes into the Early Mesozoic strata and small xenoliths of the Mesozoic strata are enclosed in the granite. The rhyolite that is spread over on top of the Manitou Formation forms an angular unconformity. The representative rock is described as follows: the granite is mainly monzogranite that is light red in color with a medium- to fine-grained subhedral texture and a blocky structure. It is mainly composed of quartz (30–35 wt%), plagioclase (25–30 wt%), potassium feldspar (30–35 wt%), and biotite (3–5 wt%). The plagioclase has a particle size of 1–4 mm and a tabular subhedral structure with of polysynthetic twinning. The potassium feldspar has a particle size of 1–6 mm and it also has a tabular subhedral structure and is mainly microcline with occasional perthite. The quartz has a particle size of 2–5 mm and mainly occurs in a subhedral granular form, and the biotite has a particle size of 1–3 mm, a sheet-like form, and noticeable brown absorbance. The accessory minerals primarily include zircon, magnetite, apatite, and titanite (Figure 2c). The rhyolite is gray or off-white and has a porphyritic texture and a weak rhyolitic structure. The porphyritic crystals are mainly plagioclase (1%–3%) and alkali feldspar (2%–6%). The plagioclase porphyritic crystals have a particle size of 2–5 mm and are in the form of plates with polysynthetic twinning. The alkali feldspar has a particle size of 1–4 mm, is also plate-like with Carlsbad twinning development and visible signs of kaolinization, and the matrix is composed of abundant felsitic felsic aggregates (Figure 2d).

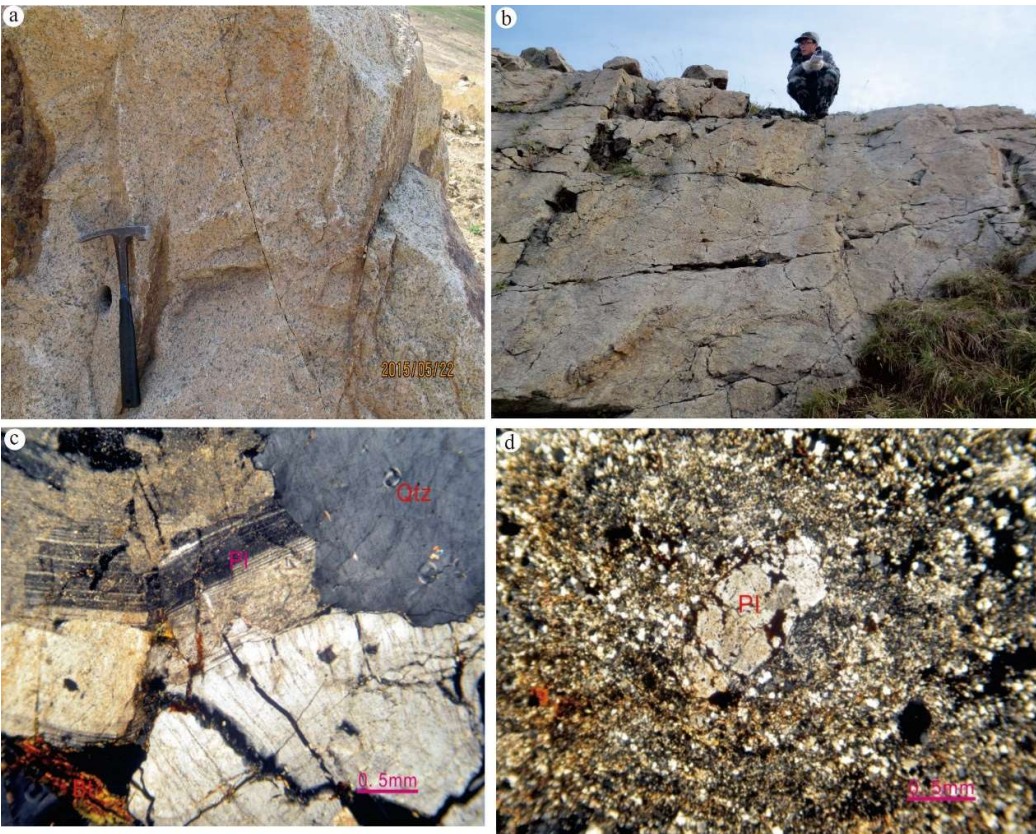

**Figure 2.** Field and microstructural photos of the Early Cretaceous magmatic rock in the Badaguan area: (**a**) granite outcrop; (**b**) rhyolite outcrop; (**c**) microscopic characteristics of the granite; (**d**) microscopic characteristics of the granite–rhyolite. Pl—plagioclase; Qtz—quartz; Bt—biotite.

## 2. Materials and Methods

Zircon sorting was performed by the Langfang Regional Geological Survey Research Institute of Hebei Province, and zircon target preparation and microscopic image acquisition were performed at the Tianjin Geological Survey Center. Zircon U-Pb chronological testing was conducted using LA-ICP-MS at the Key Laboratory of the Ministry of Land and Resources of Northeast Asia Mineral Resources Evaluation, Jilin University. The transmission, reflection, and cathodoluminescence images were acquired for the dated granite and rhyolite samples to determine the internal formation type and structural composition of zircon. High-purity helium gas was used as the carrier gas for the denudated material, and zircon U and Pb were determined using a ComPex102 ArF excimer laser at 193 nm and an Agilent 7500a ICP-MS. A synthetic silicate glass, NIST 610, developed by the American Institute of Standards and Technology, was used as a standard reference material for instrument optimization, and the Harvard zircon standard 91,500 was used for external calibration. The laser beam spot diameter was set at 30 μm for zircon determination.

The analytical data were processed by GLITTER and common Pb correction was performed according to the method proposed by [22]. Detailed experimental procedures and instrument parameters can be found in [23]. The Isoplot 3.0 program was applied for the age calculation and concordia diagram plotting. Elemental geochemical data were determined by the 240th Institute of Northeast Geological Bureau, CNNC. Major elements were tested using XRF analysis with a relative error of less than 5%, and trace and rare earth elements (REEs) were analyzed using a Perkin Elmer Elan 6100 DRC inductively coupled plasma mass spectrometer (ICP-MS) (Perkin Elmer, Waltham, MA, USA). The sample analysis was monitored using AVG-1 and BHVO-1 reference materials and the relative error was generally less than 5 wt%.

## 3. Results

### 3.1. Zircon U-Pb Chronology

The six samples tested (RZT1, RZT2, Rz9676, Rz4897, Rz4714, and Rz4773) all had short or long prismatic shapes. They also displayed oscillatory zoning and high Th/U ratios (0.54–2.86), thereby revealing the characteristics of magmatic zircon [24]. The analytical results are shown in Table S1. The results of the sample analyses are described as follows.

For sample RZT1 (granite), all 20 of the zircon samples tested were located on or near the U-Pb concordia line. The $^{206}$Pb–$^{238}$U apparent age ranged from 125 to 145 Ma and the weighted average age was 132 ± 3 Ma with a mean squared weight deviation (MSWD) of 3.3, which indicates that the granite was formed in the Early Cretaceous period (Figure 3a).

For sample RZT2 (granite), the 20 zircon samples tested were also all located on or near the U-Pb concordia line. The $^{206}$Pb–$^{238}$U apparent ages ranged from 121 to 151 Ma with a weighted average age of 135 ± 3 Ma and a MSWD of 2.3, which indicates that the granite was also formed in the Early Cretaceous period (Figure 3b).

For sample Rz9676 (rhyolite), 19 out of the 20 zircon samples tested were located on or near the U-Pb concordia line, with the $^{206}$Pb–$^{238}$U apparent ages ranging from 131 to 148 Ma, a weighted average age of 140 ± 2 Ma, and a MSWD of 1.7, which indicates that the rhyolite was also formed in the Early Cretaceous period. However, one zircon point had the relatively young age of 112 ± 3 Ma, and this is potentially attributed to local tectono-thermal events (Figure 3c).

For sample Rz4897 (rhyolite), 18 of the 20 zircon samples tested were located on or near the U-Pb concordia line. The $^{206}$Pb–$^{238}$U apparent ages ranged from 122 to 145 Ma with a weighted average age of 137 ± 3 Ma and a MSWD of 2.3, which also indicates that the rhyolite was formed in the Early Cretaceous period. However, the two remaining points were dated at 218 ± 2 Ma and 158 ± 2 Ma, which implies that they were captured zircon (Figure 3d).

For sample Rz4714 (rhyolite), 17 of the 20 zircon samples analyzed were located on or near the U-Pb concordia line, with $^{206}$Pb–$^{238}$U apparent ages ranging from 120 to 138 Ma.

The weighted average age was 137 ± 3 Ma and the MSWD was 2.3, which also indicates that the rhyolite was formed in the Early Cretaceous period. However, the other three zircon samples were dated at 151 ± 2 Ma, 151 ± 2 Ma, and 155 ± 2 Ma, which implies that they were potentially captured zircon (Figure 3e).

For sample Rz4773 (rhyolite), 19 of the 20 zircon samples tested were located on or near the U-Pb concordia line. The $^{206}$Pb–$^{238}$U apparent ages ranged from 115 to 135 Ma, the weighted average age was 125 ± 3 Ma, and the MSWD was 2.7, which indicates that the rhyolite was also formed in the Early Cretaceous period. The remaining zircon points were older with ages of 216 ± 2 Ma, which also implies that they were potentially derived from assimilated earlier rock zircon (Figure 3f).

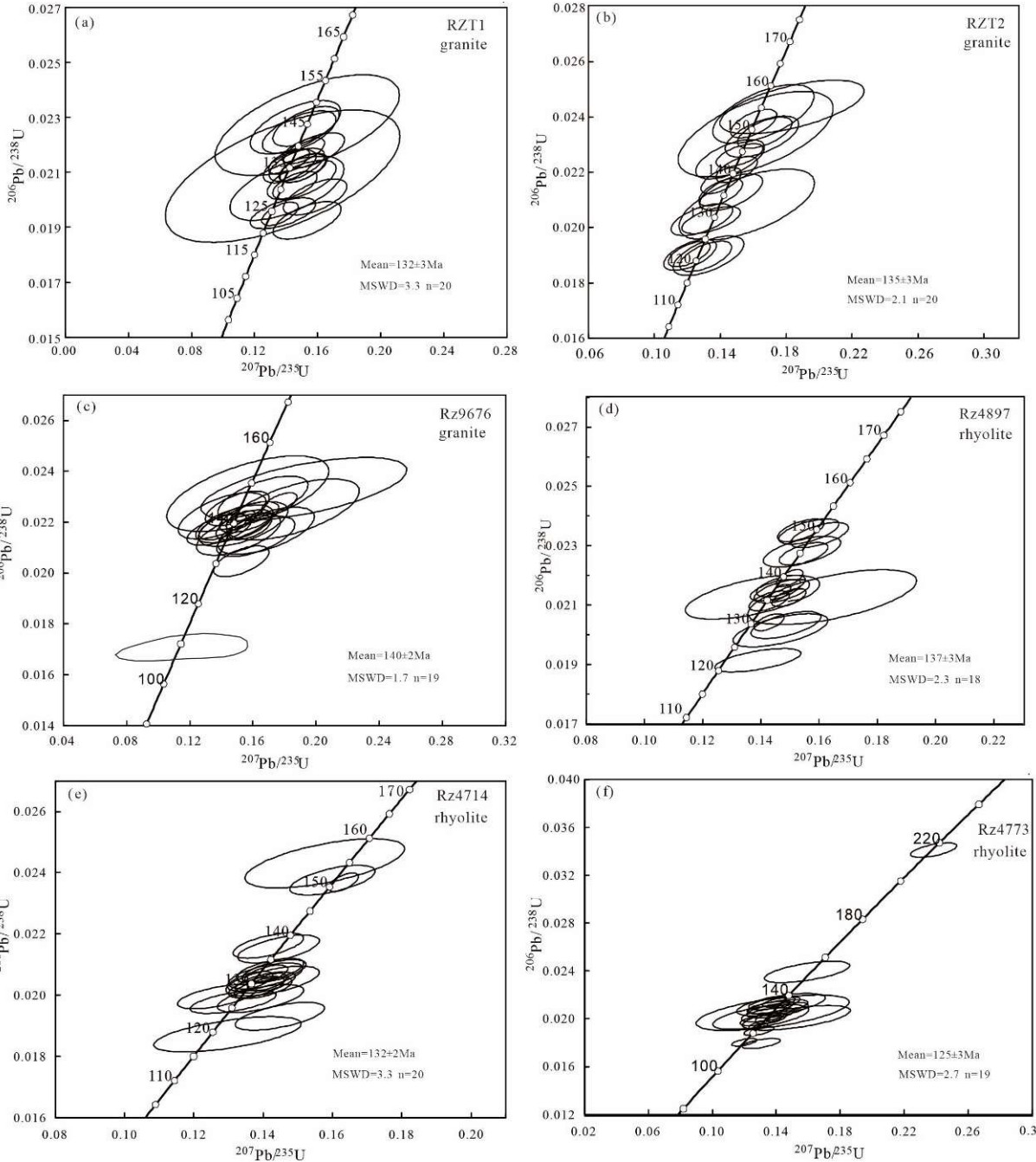

**Figure 3.** Concordia diagrams of the Early Cretaceous granite and rhyolite from the Badaguan area. (**a–c**) are granites; (**d–f**) are rhyolites.

### 3.2. Whole-Rock Geochemical Characteristics

### 3.2.1. Major Elements

The samples of Early Cretaceous magmatic rock from the Badaguan area had a high $SiO_2$ content, ranging from 61.68% to 77.41 wt% with an average value of 71.22 wt%. The total alkali ($Na_2O + K_2O$) content was from 5.16 wt% to 9.33 wt% with an average value of 6.63 wt%. The rock also had a high aluminum content ($Al_2O_3$ ranged from 10.64 wt% to 16.94 wt% with an average value of 13.98 wt%) and low magnesium and calcium contents (MgO ranged from 0.15 wt% to 1.66 wt% with an average of 0.69 wt%; CaO ranged from 0.29 wt% to 4.63 wt% with an average of 1.44 wt%). The A/CNK ratio was between 0.99 and 1.58 with an average of 1.22 (Table S2), which implies a weakly peraluminous to peraluminous nature (Figure 4a). With respect to the $SiO_2$–$K_2O$ plot, most of the rhyolite falls into the high-potassium, calc-alkaline region, while all of the granite is gathered within the calc-alkaline region (Figure 4b). Their characteristics were fairly similar to those of the I-type granite detected in Northeast China [25–28].

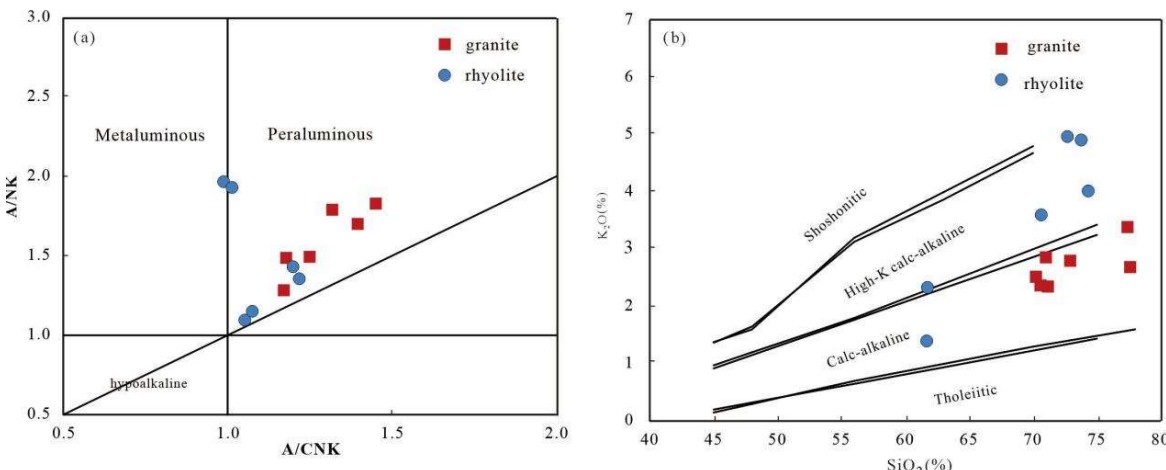

**Figure 4.** (**a**) A/NK vs. A/CNK, diagrams of the Early Cretaceous magmatic rock from the Badaguan area, modified from Maniar and Piccoli. [29] and (**b**) $SiO_2$ vs. $K_2O$ diagrams of the Early Cretaceous magmatic rock from the Badaguan area, modified from Peccerllo and Taylor [30].

### 3.2.2. Trace and Rare Earth Elements (REEs)

The samples of the Early Cretaceous magmatic rock in the Badaguan area contained a slightly higher than an average level of total REEs (ΣREE is in the range of 99.78 ppm−301.05 ppm with an average value of 154.9 ppm). The REE distribution pattern was right-inclined overall (Figure 5a), with relatively enriched LREEs and depleted HREEs. The light and heavy REE ratios ranged from 4.37 to 11.68, which is characteristic of crustal origin rock [31]. The fractionation coefficients ranged from 1.42 to 4.59 for the light REEs $(La/Sm)_N$ and from 0.88 to 2.98 for the HREEs $(Gd/Yb)_N$, with a more substantial trend of LREE fractionation than HREE fractionation. A weak Eu negative anomaly (δEu in the range of 0.21−0.93) was detected and the negative Eu anomaly of the granite was more noticeable through a lithological comparison of the two types. The primitive mantle normalized spider diagram (Figure 5b) shows that all samples exhibited a consistent evolutionary trend that was characterized by the enrichment of large-ion lithophile elements (LILEs, Rb, Th, U), strongly depleted Sr, P, and Ti, and the relative depletion of Nb (Table S2).

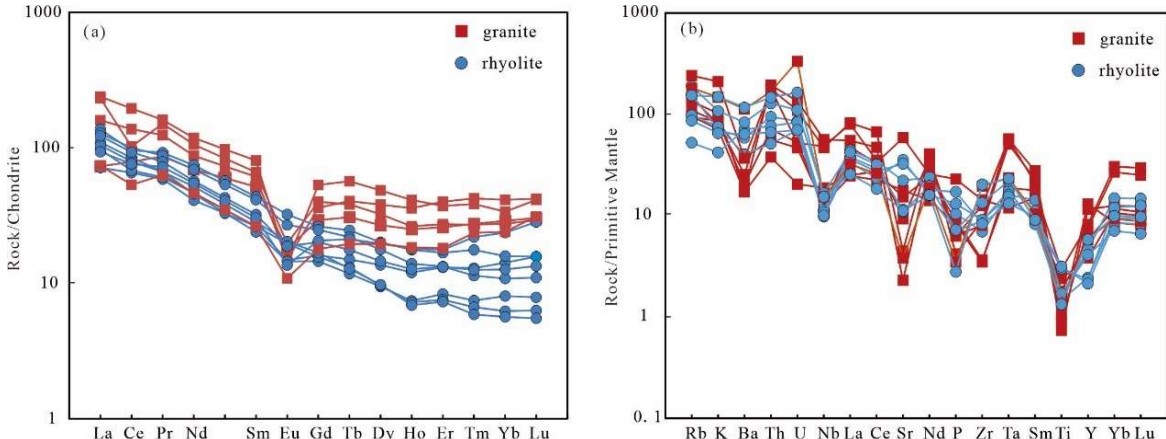

**Figure 5.** (**a**) The chondrite normalized REE pattens for the Early Cretaceous magmatic rock from the Badaguan area, modified from Boynton [32] and (**b**) the primitive mantle normalized trace element spider diagrams for the Early Cretaceous magmatic rock from the Badaguan area, modified from Sun and McDonough [33].

## 4. Discussion

### 4.1. Age of Magmatic Rock Formation

Due to the lack of precise chronological and paleontological data, the formation of the granite was determined to have taken place during the Hercynian period and the rhyolite was concluded to have been formed in the Late Jurassic (i.e., the Manketou'ebo Formation) using stratigraphic contact and comprehensive comparison. However, this study aimed to accurately determine the formation age of the magmatic rock in this region using high-accuracy zircon U-Pb dating through LA-ICP-MS. The U-Pb ages of the Hercynian granite and the rhyolite in the Manketou'ebo Formation were found to differ from those previously concluded; they were determined to be 132–140 Ma and 125–137 Ma, respectively, both of which relate to the Early Cretaceous period, which is later than the previously assigned Indosinian and Late Jurassic periods.

Detailed chronological data obtained in recent years have shown that Early Cretaceous magmatism was fairly intense above the Erguna Massif; some examples of assemblages that evidence this magmatism are (1) the basalt and basaltic andesite assemblage in the Yakeshi area (114–116 Ma), (2) the ferromagnesian and feldspathic rock assemblage in the Shanghulin area (114–124 Ma), (3) the trachyte and rhyolite assemblages in the Lingquan Basin (125–141 Ma), and (4) the basalt and trachyte assemblages in the Hailar Basin (122–134 Ma) (Table S3 [34–38]). By combining the results of previous studies with the results of this paper, it is clear that the entire Erguna Massif (including the study area) experienced an important magmatic event in the Early Cretaceous period.

### 4.2. Magma Source Characteristics and Petrogenesis

The sample of Early Cretaceous magmatic rock from the study area comprised mainly granites and rhyolites, and their mineral composition was dominated by potassium feldspar, plagioclase, and quartz with small amounts of biotite. However, they lacked aluminum-rich minerals, such as cordierite and garnet. The characteristics were similar to those of metaluminous–weakly peraluminous rocks and the $P_2O_5$ and $Al_2O_3$ contents exhibited substantial negative correlations with $SiO_2$. The comprehensive whole-rock geochemical analyses of the magmatic rock samples indicated that the Early Cretaceous magmatic rock in the study area lacks the features of S-type granite [39] and can instead be tentatively classified as I-type or A-type. Almost all of the magmatic rock samples fell within the areas of I-type or highly differentiated I-type granites in the discrimination diagrams of granite formation (Figure 6a,b). Therefore, the Early Cretaceous magmatic rock of the Badaguan region was determined as being I-type.

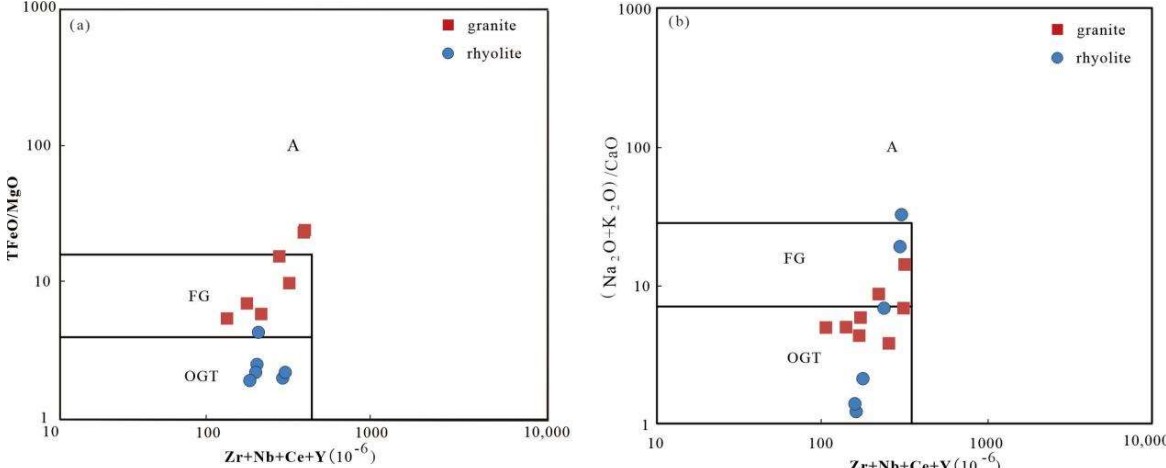

**Figure 6.** A discrimination diagram of the Early Cretaceous magmatic rock from the Badaguan area. (**a**)TFeO/MgO–Zr + Nb + Ce + Y diagrams; (**b**) (Na$_2$O + K$_2$O)/CaO–Zr + Nb + Ce + Y diagrams.

Table S3 shows the zircon U-Pb ages of the samples of Early Cretaceous magmatic rock from the Erguna Massif. The results of the trace element geochemical analyses show that the magmatic rock had Th/U ratios of between 2.04 and 13.94 (with an average of 5.34) and that these ratios were close to that of the lower crust (6.00). The Rb/Sr ratio (ranging from 0.04 to 1.77 with an average of 0.54) also fell within the range of crustal source magma [40] and was distant from the Rb/Sr values [33] of the primitive mantle (0.03), OIB (0.047), and E-MORB (0.033). The Nb/Ta ratio was between 9.01 and 18.08 (with an average value of 13.57), which was lower than the depleted mantle ratio (>17) but higher than the lower crust (8.3). Furthermore, the significant depletion of P and Ti and the low contents of compatible elements (such as Cr, Co, and Ni) also reflected the fact that the magma had a crustal source rather than a mantle source.

Previously published data from the Erguna Block have also shown that the zircon has a relatively homogeneous Hf isotopic composition (Table S4). The $\varepsilon$Hf(t) values in the rock are positive overall (+1 ~ 11.9) and the model age is relatively young, with a T$_{DM2}$ within the range of 0.31–1.15 Ga, which is consistent with the $\varepsilon$Hf(t) values of the vastly exposed Phanerozoic granites in the Xingmeng orogenic belt [41–47]. This implies that the whole Xingmeng orogenic belt, including the Early Cretaceous magmatic rock in the Badaguan area, has a similar magma source and that the source rock was formed by the partial melting of newly generated basal crustal material from the depleted mantle during the Meso-Neoproterozoic period.

The Early Cretaceous magmatic rock in the study area has a high SiO$_2$ content and more basic rock of the same period has been identified on the Erguna Massif [18]. Therefore, the negative Eu anomalies and strong Sr depletion in the samples of Early Cretaceous magmatic rock from the Badaguan area could be due to the occurrence of significant fractional crystallization processes in the magma. In addition, on the Harker diagram (Figure 7), there are noticeable linear variations between all of the magmatic rock samples, and the MgO, Fe$_2$O$_3$$^T$, and CaO values display substantial negative correlations with that of SiO$_2$. This indicates that the considerable fractional crystallization of plagioclase and hornblende occurred in the magma during its evolution, and this speculation is also confirmed by the fractional crystallization vector diagram (Figure 8). Furthermore, the magmatic rock samples had relatively low (La/Yb)$_N$ and (Gd/Yb)$_N$ ratios but were enriched with HREEs, which suggests that the source region was almost garnet-free [42]. It is also evident from the Dy/Yb–Gd/Yb diagram that the melting point of magmatic rock samples was close to the spinel–peridotite melting point but far from the garnet–peridotite melting point (Figure 8a). The TiO$_2$, P$_2$O$_5$, and SiO$_2$ were significantly negatively correlated. Along with the strong depletion of P and Ti in the trace element diagram, this implies

that the magma source area also experienced the fractional crystallization of apatite and Fe–Ti oxides.

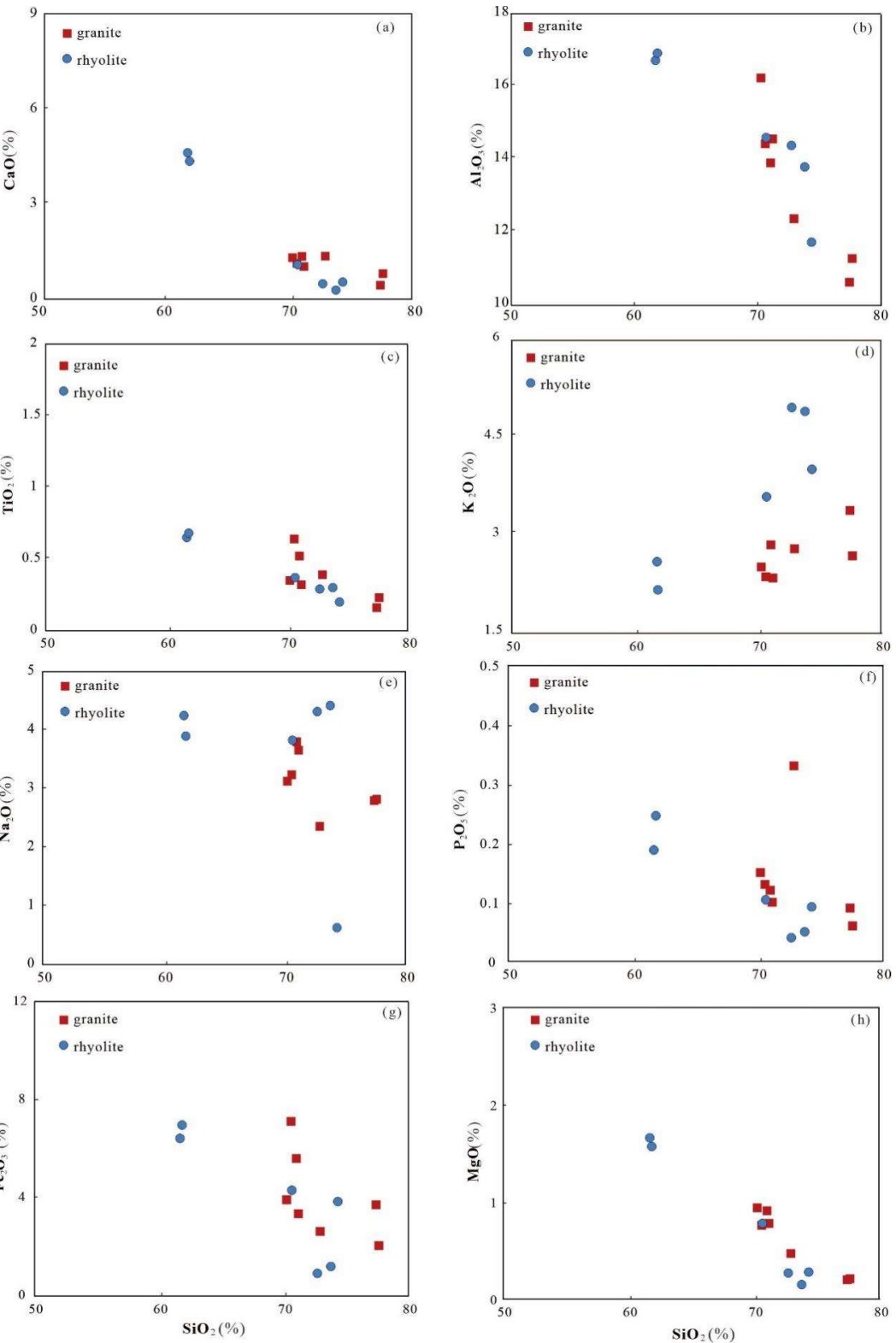

**Figure 7.** Harker diagram of the Early Cretaceous magmatic rock from the Badaguan area. (**a**): CaO–SiO₂ diagrams; (**b**): Al₂O₃–SiO₂ diagrams; (**c**): TiO₂–SiO₂ diagrams; (**d**): K₂O–SiO₂ diagrams; (**e**): Na₂O–SiO₂ diagrams; (**f**): P₂O₅–SiO₂ diagrams; (**g**): Fe₂O₃ᵀ–SiO₂ diagrams; (**h**): MgO–SiO₂ diagrams.

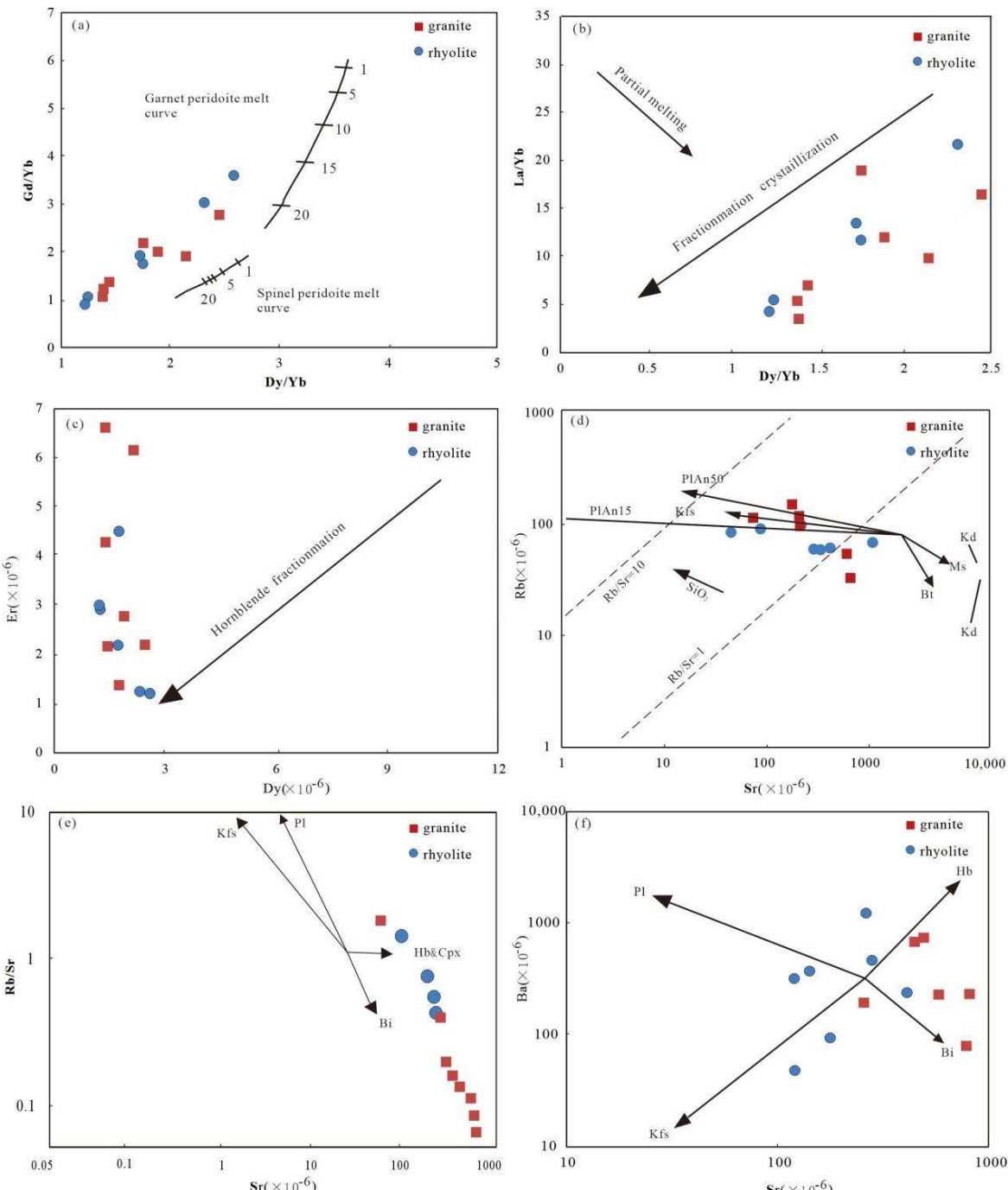

**Figure 8.** A fractional crystallization diagram of the Early Cretaceous magmatic rock from the Badaguan area. (**a**): Gd/Yb–Dy/Yb diagrams; (**b**): La/Yb–Dy/Yb diagrams; (**c**): Er–Dy diagrams; (**d**): Rb–Sr diagrams; (**e**): Rb/Sr–Sr diagrams; (**f**): Ba–Sr diagrams.

In summary, the Early Cretaceous magmatic rock samples investigated in this study displayed the characteristics of I-Type granite and the results imply that the magma source was the partial melting of basal crustal material that was newly generated from the depleted mantle during the Meso-Neoproterozoic period. The substantial fractional crystallization of plagioclase, hornblende, apatite, and other minerals occurred during the evolution of the magma and garnet was almost non-existent in the residual phase within the source area.

### 4.3. Tectonic Background

It was traditionally agreed that the entire northeast region of China was in an extensional tectonic setting during the Early Cretaceous period [18,37,43]. For example, the metamorphic core complex belt in the Daxing'anling area spreads in a northeasterly direction, which aligns with the spreading direction of the Mongol-Okhotsk Ocean (which has a formation age of 130–150 Ma) [10,36]. In addition, the previously discovered bimodal volcanic rock assemblages, A-type granitic rock, and alkaline rhyolites were all formed in the Early Cretaceous age, which further confirms that the Erguna Massif was in an extensional environment during the Early Cretaceous period [48,49].

The extensional environment in the Early Cretaceous period is currently being debated, and the main points of views are as follows: (1) it was related to the post-closure extensional environment of the Paleo-Asian Ocean; (2) it was linked to the extensional environment of the Paleo-Pacific Ocean [13,50]; and (3) it was associated with the post-closure extensional environment of the Mongol-Okhotsk Ocean [14,37,38]. Northeast China was mainly under the dual influence of the Paleo-Asian Ocean and the Mongol-Okhotsk Ocean during the Late Permian–Early Triassic period [15,34], while the Paleo-Asian Ocean closed along the Solonker–Xar Moron River–Changchun–Yanji suture zone in the Late Permian–Early Triassic period, forming a large-scale magmatic belt trending nearly east–west on the Songnen Massif [18]. Bimodal volcanic rock and A-type high-potassium, calc-alkaline volcanic rock is mainly associated with the post-collisional extensional environment [18]. However, this suture zone is predominantly located in the southern part of the Daxing'anling region, which is located approximately 1000 km away from the study area. It has a minimal influence on the area, both spatially and temporally. In addition, previous chronological data have shown that two important geological periods existed in Northeast China: the Late Jurassic (161–147 Ma) period and the Early Cretaceous period (106–141 Ma). Most of the Late Jurassic magmatic rock was formed primarily in the western part of the Songliao Basin [35], which is dominated by high-potassium, calc-alkaline rock [38]. Assuming that the tectonic environment was influenced by the subduction of the Paleo-Pacific Plate at that time, a large amount of low-potassium tholeiitic basalt and high-potassium, calc-alkaline magmatic rock would have developed around eastern Jilin, Heilongjiang, and the Songliao Basin. However, such rock has not been discovered in these areas. Furthermore, the spatial and temporal distribution of the Early Cretaceous magmatic rock around the Mongol-Okhotsk Ocean suggests that the magmatism was associated with the construction system of the Mongol-Okhotsk Ocean instead of the Paleo-Pacific tectonic system [51–53]. Furthermore, the characteristics of the granite and adakitic intrusive rock discovered in the northwestern part of the Erguna Massif indicate that the Mongol-Okhotsk Ocean closed in the Middle Jurassic period [7]. This led to the interaction between the Siberian Craton and the Central Mongolian Massif and the transition of the tectonic system from crustal thickening to extensional thinning during the Late Jurassic period. A series of A-type felsic rock was formed in the northern part of the Daxing'anling Mountains during the Late Jurassic–Early Cretaceous period, which suggests that there was a connection between the spread of the postnatal lithosphere and the closure of the Mongol-Okhotsk Ocean [18].

With respect to the structure diagrams (Figure 9), most of the magmatic rock samples from the study area were gathered in the volcanic arc granite zone, which suggests a close relationship with the lithospheric extensional environment after the closure of the Mongol-Okhotsk Ocean. In addition, the previous discovery of Early Cretaceous A-type granite and alkaline rhyolite [35,36], bimodal felsic rock [49], and the metamorphic core complex zone [2] in the Erguna Massif indicates that the Early Cretaceous magmatism of the Erguna Block was associated with the post-collisional extension of the Mongol-Okhotsk belt. However, Early Cretaceous magmatic rock appeared not only in and around the Badaguan area of the Erguna Massif, but also in Transbaikal and on the Sino-Russian border [54,55] and thus, they geographically span the entire Mongol-Okhotsk Ocean suture zone. Therefore, the regions under the influence of the Mongol-Okhotsk Ocean and the Paleo-Pacific Ocean could be delineated by the Songliao Basin, where magmatism in the

central and eastern part of the basin is mainly related to the subduction of the Paleo-Pacific plate, while magmatism in the western part of the basin (i.e., northern Daxing'anling) is primarily linked to the post-orogenic extension of the Mongol-Okhotsk orogenic belt [18]. Furthermore, the Erguna Massif geographically lies at a certain distance from the East Asian Continental Margin, and seismic data have shown that the back arc extension of the Paleo-Pacific Plate did not reach the area to the west of the Songliao Basin. Thus, it is evident that the subduction of the Paleo-Pacific Plate had a minimal influence on the Early Cretaceous magmatism of the Erguna Massif.

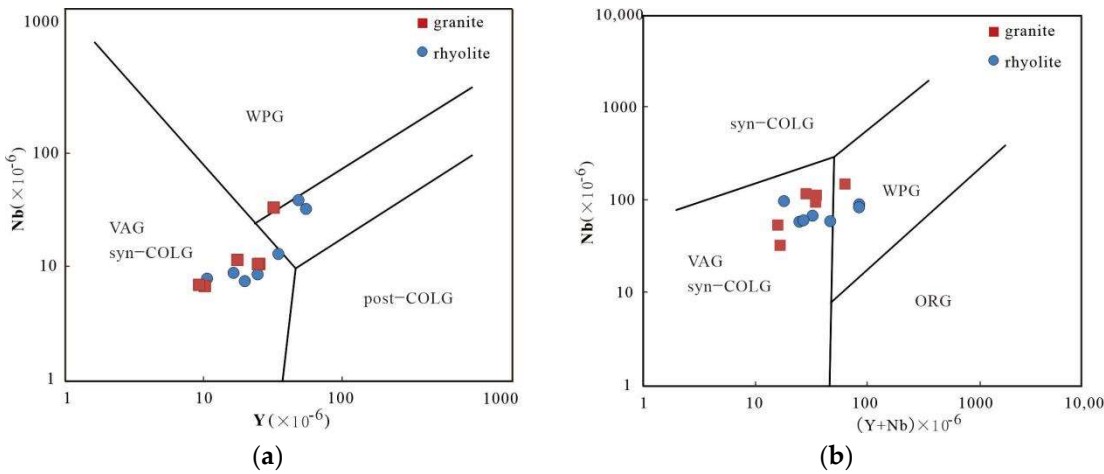

**Figure 9.** A structure diagram of the Early Cretaceous magmatic rock from the Badaguan area. (**a**): Nb–Y diagrams; (**b**): Nb–Y + Nb diagrams.

## 5. Conclusions

Based on the chronology and geochemistry of the magmatic rock in the Badaguan area combined with existing research, we drew the following main conclusions:

(1) The magmatic rock in the Badaguan area in the northern part of the Daxing'anling Mountains is mainly composed of granite and rhyolite. Contrary to previous conclusions, it was formed between 125 to 140 Ma rather than during the Indosinian and Late Jurassic periods;

(2) The origins of the Early Cretaceous magmatic rock in the Badaguan area of the northern Daxing'anling Mountains are characterized by crustal source magma and I-type granite. They evolved from the partial melting of newly generated basal crustal materials from the depleted mantle during the Meso-Neoproterozoic period. The substantial fractional crystallization of plagioclase, hornblende, apatite, and other minerals occurred during the magma evolution process. The residual phase of the source area contained almost no garnet;

(3) The Early Cretaceous magmatic rock in the Badaguan area recorded the lithospheric extensional environment after the closure of the Mongol-Okhotsk Ocean, and this is consistent with the results of previous studies that have investigated synchronous tectono-magmatic activities in Northeast China.

**Supplementary Materials:** The following supporting information can be downloaded at: https://www.mdpi.com/article/10.3390/min12030303/s1, Table S1: LA-ICP-MS zircon U-Pb dating of Early Cretaceous magmatic rocks in Badaguan area; Table S2: Analysis of main elements, trace elements and REE of Early Early Cretaceous magmatic rocks in Badaguan area; Table S3: Zircon U-Pb dating of the Early Cretaceous magmatic rocks in the Erguna block; Table S4: Hf isotopic characteristics of the Early Cretaceous magmatic rocks in the Erguna block.

**Author Contributions:** Y.L., J.-R.C., Z.-B.H., F.-J.N., X.-G.H. and Z.-B.Y. designed the project; Y.L. wrote and organized the paper, with a careful discussion and revision by J.-R.C., Z.-B.H., F.-J.N., X.-G.H. and Z.-B.Y. All authors have read and agreed to the published version of the manuscript.

**Funding:** This work was financially supported by the China Geological Survey (grant no. 12120113053700) and the Fundamental Science on Radioactive Geology and Exploration Technology Laboratory, East China University of Technology (grant no. 2020RGET02).

**Data Availability Statement:** The data presented in this study are available in this article.

**Conflicts of Interest:** The authors declare no conflict of interest.

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
