# Peer review of "Chronology and Geochemistry of Early Cretaceous Magmatism in the Northwestern Erguna Block, Northeast China"

_minerals, doi:10.3390/min12030303_

Round 1

Reviewer 1 Report

- The geological map of the studied area should contain also the positioning of the surveyed area at the country level, respectively China;

- The tables with the values of the performed analyses (Zircon U-Pb chronology, Major elements and Trace and rare earth elements (REEs) are missing. In the text there is reference to the tables 1,2, 3 and 4, but in the folder I had received the aforementioned tables do not exist;

- At page , paragraph 1, a phrase is repeating, i.e. : „ However, this study aimed to accurately determine 213 the formation age of the magmatic rocks in this region using high-accuracy, zircon U-Pb 214 dating through LA-ICP-MS.

Author Response

The geological map of the studied area should contain also the positioning of the surveyed area at the country level, respectively China

Response: We have added

The tables with the values of the performed analyses (Zircon U-Pb chronologyMajor elements and Trace and rare earth elements (REEs) are missing. In the text there is reference to the tables 1,2, 3 and 4, but in the folder I had received the aforementioned tables do not exist;

Response: We have submitted to the editorial board in the first round of review

At page , paragraph 1, a phrase is repeating, i.e. : „ However, this study aimed to accurately determine 213 the formation age of the magmatic rocks in this region using high-accuracy, zircon U-Pb 214 dating through LA-ICP-MS.

Response: We have deleted

Reviewer 2 Report

The following will improve the paper

Line 11 Delete  Zircon U-Pb dating.  Replace with  These results  [The dating does not identify the rock types & is a repetition of the proceeding line]

Line 12 Delete    . The results of the geochemical analyses display that they   Replace with  , and      {have a.......}

Line 14 Full stop after heavy REEs. When combined........

Line 15 Insert   this suggests

Line 19 extensional

Line 20 hypothesis

Line 31 Delete activities & replace magmatic, with magmatism,

Line 36 follows (Figure 1b):      ie insert a reference to the figure

Line 54 insert a space after gneisses but also on pages 27, 39, 43, 58, 61, 66, 68 & elsewhere, go through & put a space which is at present inconsistently missing.

Line 89 & 234 Why is 'black mica' not named as usual, biotite? 

Line 90 'that shows evidence of' replace with    with

Line 99 Delete developed

Line 105 outcrope , delete e; Replace 2 Mirrors by Microscopic

line 106 Insert hyphen granite-rhyolite

Line 135 replace columnar with prismatic

Line 168 Delete captured & replace it with  derived from assimilated earlier rocks

Line 170 Concordia diagrams of Early Cretaceous granites in the Budaguan area.

Line 213-215 & 215-217 are repetitions. Delete one sentence.

Line 220 earlier is wrong; Cretaceous is later so replace earlier with later

Line 270 & 284 Harker not Haker

Fig 7(e) Very curious to have ~0.5% K20 in a rhyolite with 74% SiO2 CHECK?

Author Response

Line 11 Delete  Zircon U-Pb dating.  Replace with  These results  [The dating does not identify the rock types & is a repetition of the proceeding line]

Response: We have replaced

Line 12 Delete    . The results of the geochemical analyses display that they   Replace with  , and      {have a.......}

Response: We have replaced

Line 14 Full stop after heavy REEs. When combined........

Response: We have added

Line 15 Insert   this suggests

Response: We have added

Line 19 extensional

Response: We have modified

Line 20 hypothesis

Response: We have modified

Line 31 Delete activities & replace magmatic, with magmatism,

Response: We have modified

Line 36 follows (Figure 1b):      ie insert a reference to the figure

Response: We have modified

Line 54 insert a space after gneisses but also on pages 27, 39, 43, 58, 61, 66, 68 & elsewhere, go through & put a space which is at present inconsistently missing.

Response: We have modified

Line 89 & 234 Why is 'black mica' not named as usual, biotite? 

Response: We have modified

Line 90 'that shows evidence of' replace with    with

Response: We have modified

Line 99 Delete developed

Response: We have modified

Line 105 outcrope , delete e; Replace 2 Mirrors by Microscopic

Response: We have modified

line 106 Insert hyphen granite-rhyolite

Response: We have modified

Line 135 replace columnar with prismatic

Response: We have modified

Line 168 Delete captured & replace it with  derived from assimilated earlier rocks

Response: We have modified

Line 170 Concordia diagrams of Early Cretaceous granites in the Budaguan area.

Response: We have modified

Line 213-215 & 215-217 are repetitions. Delete one sentence.

Response: We have modified

Line 220 earlier is wrong; Cretaceous is later so replace earlier with later

Response: We have modified

Line 270 & 284 Harker not Haker

Response: We have modified

Fig 7(e) Very curious to have ~0.5% K20 in a rhyolite with 74% SiO2 CHECK?

Response: We have modified
